# COVID-19 epidemiology and changes in health service utilization in Azraq and Zaatari refugee camps in Jordan: A retrospective cohort study

**Chiara Altare**[1,2], **Natalya Kostandova**[1,2], **Jennifer OKeeffe**[1,2], **Heba Hayek**[3], **Muhammad Fawad**[3], **Adam Musa Khalifa**[3], **Paul B. Spiegel**[1,2]*

1 Johns Hopkins Bloomberg School of Public Health, Baltimore, Maryland, United States of America, 2 Johns Hopkins Center for Humanitarian Health, Baltimore, Maryland, United States of America, 3 United Nations High Commissioner for Refugees, Amman, Jordan

* pbspiegel@jhu.edu

**Data Availability Statement:** Anonymized and de-identified individual-level data (COVID-19) and aggregated weekly data (routine health services)

## Abstract

### Background

The effects of the Coronavirus Disease 2019 (COVID-19) pandemic in humanitarian contexts are not well understood. Specific vulnerabilities in such settings raised concerns about the ability to respond and maintain essential health services. This study describes the epidemiology of COVID-19 in Azraq and Zaatari refugee camps in Jordan (population: 37,932 and 79,034, respectively) and evaluates changes in routine health services during the COVID-19 pandemic.

### Methods and findings

We calculate the descriptive statistics of COVID-19 cases in the United Nations High Commissioner for Refugees (UNHCR)'s linelist and adjusted odds ratios (aORs) for selected outcomes. We evaluate the changes in health services using monthly routine data from UNHCR's health information system (HIS; January 2018 to March 2021) and apply interrupted time series analysis with a generalized additive model and negative binomial (NB) distribution, accounting for long-term trends and seasonality, reporting results as incidence rate ratios (IRRs).

COVID-19 cases were first reported on September 8 and September 13, 2020 in Azraq and Zaatari camps, respectively, 6 months after the first case in Jordan. Incidence rates (IRs) were lower in camps than neighboring governorates (by 37.6% in Azraq (IRR: 0.624, 95% confidence interval [CI]: [0.584 to 0.666], p-value: <0.001) and 40.2% in Zaatari (IRR: 0.598, 95% CI: [0.570, 0.629], p-value: <0.001)) and lower than Jordan (by 59.7% in Azraq (IRR: 0.403, 95% CI: [0.378 to 0.430], p-value: <0.001) and by 63.3% in Zaatari (IRR: 0.367, 95% CI: [0.350 to 0.385], p-value: <0.001)). Characteristics of cases and risk factors for negative disease outcomes were consistent with increasing COVID-19 evidence. The following health services reported an immediate decline during the first year of COVID-19: healthcare

may be made available upon request following publication and approval of a concept note summarizing the analyses to be conducted. Please contact Johns Hopkins' Center for Humanitarian Health at humanithealth@jhu.edu.

**Funding:** This work was supported by US Centers for Disease Control and Prevention, grant number 5NU2GGH002000-04-00 (CA, PS, NK, JOK). The funders had no role in study design, data collection, analysis, interpretation, writing of the manuscript, or decision to publish.

**Competing interests:** I have read the journal's policy and the authors of this manuscript have the following competing interests: MK, FH and HH are employed by UNHCR. CA, PS, NK and JOK declare no competing interests.

**Abbreviations:** ANC, antenatal care; ANOVA, analysis of variance; aOR, adjusted odds ratio; CI, confidence interval; COVID-19, Coronavirus Disease 2019; HIS, health information system; ILI, influenza-like illness; IR, incidence rate; IRB, Institutional Review Board; IRR, incidence rate ratio; LRTI, lower respiratory tract infection; NB, negative binomial; NCD, noncommunicable disease; NPI, nonpharmaceutical intervention; RTI, respiratory tract infection; SARS-CoV-2, Severe Acute Respiratory Syndrome Coronavirus 2; STROBE, Strengthening the Report of Observational Studies in Epidemiology; UNHCR, United Nations High Commissioner for Refugees; URTI, upper respiratory tract infection.

utilization (by 32% in Azraq (IRR: 0.680, 95% CI [0.549 to 0.843], $p$-value < 0.001) and by 24.2% in Zaatari (IRR: 0.758, 95% CI [0.577 to 0.995], $p$-value = 0.046)); consultations for respiratory tract infections (RTIs; by 25.1% in Azraq (IRR: 0.749, 95% CI: [0.596 to 0.940], $p$-value = 0.013 and by 37.5% in Zaatari (IRR: 0.625, 95% CI: [0.461 to 0.849], $p$-value = 0.003)); and family planning (new and repeat family planning consultations decreased by 47.4% in Azraq (IRR: 0.526, 95% CI: [0.376 to 0.736], $p$-value = <0.001) and 47.6% in Zaatari (IRR: 0.524, 95% CI: [0.312 to 0.878], $p$-value = 0.014)). Maternal and child health services as well as noncommunicable diseases did not show major changes compared to pre–COVID-19 period. Conducting interrupted time series analyses in volatile settings such refugee camps can be challenging as it may be difficult to meet some analytical assumptions and to mitigate threats to validity. The main limitation of this study relates therefore to possible unmeasured confounding.

## Conclusions

COVID-19 transmission was lower in camps than outside of camps. Refugees may have been affected from external transmission, rather than driving it. Various types of health services were affected differently, but disruptions appear to have been limited in the 2 camps compared to other noncamp settings. These insights into Jordan's refugee camps during the first year of the COVID-19 pandemic set the stage for follow-up research to investigate how infection susceptibility evolved over time, as well as which mitigation strategies were more successful and accepted.

## Author summary

### Why was this study done?

- There is a scarcity of information on the Coronavirus Disease 2019 (COVID-19) in humanitarian settings, including in refugee camps.

- Challenges specific to humanitarian settings have raised concerns over the ability to respond to the pandemic as well as to maintain essential health services.

- Crises often result in diverted attention and funding for health services that are critical to preventing excess disease and death from all causes.

- Evidence about the COVID-19 situation in humanitarian settings can help actors to make informed decisions about epidemic response and to appropriately prioritize services.

### What did the researchers do and find?

- We used observational programmatic data to describe the COVID-19 situation in 2 refugee camps in Jordan (Azraq and Zaatari camps) and evaluated changes in health services pre- and during COVID-19 periods.

- We found that there were lower rates of COVID-19 in the camps than at the governorate level by 37.6% (Azraq) and 40.2% (Zaatari) and at the national level by 59.7% (Azraq) and 63.3% (Zaatari).

- At the beginning of the COVID-19 pandemic, health service utilization declined for health consultations by 32.0% (Azraq) and 24.2% (Zaatari), for respiratory tract infection (RTI) consultations by 25.1% (Azraq) and 37.5% (Zaatari), and for family planning services by 47.4% (Azraq) and 47.6% (Zaatari).

- Health services for maternal healthcare and noncommunicable diseases (NCDs) did not show major changes between the pre- and during COVID-19 periods.

## What do these findings mean?

- The findings indicate that refugees did not pose a threat of spreading COVID-19 in the study locations, but may themselves have been affected by external COVID-19 transmission.

- There was a mix of health service performance in the first year of the pandemic, although services in the camps appeared to function better than in noncamp settings.

- Findings are limited by the challenges to conducting research in humanitarian settings including difficulties measuring health system performance and capturing external characteristics that could affect study results.

- The results set the stage for follow-up research on COVID-19 in humanitarian settings, including how infection rates may change over time, which mitigation strategies are appropriate, and how health services are affected.

## Introduction

The novel Coronavirus Disease 2019 (COVID-19) was first detected in China in December 2019 [1]. The transmission pace and dynamics have varied widely, with multiple waves of cases affecting countries at different times and magnitudes. The effects of and response to the pandemic in humanitarian settings are not well understood as data are limited and of poor quality. The number of cases in such settings has increased over time [2], yet reported cases and deaths have not reached high levels of other countries like Brazil or India [2], nor the gloomy scenarios from initial modeling exercises [3,4].

Vulnerabilities specific to humanitarian contexts include precarious living conditions, poor access to water, overcrowding, lack of cleaning supplies, and dependence upon external funding [5,6]. These factors raised concerns about governments' and the international community's capacities to respond to the pandemic and to maintain essential health services [7]. Furthermore, diverting funding and attention to the pandemic may increase vulnerability to and negative outcomes from other diseases. In previous large scale epidemics (e.g., Ebola in West Africa and cholera in Yemen), there was excess morbidity and mortality from communicable and noncommunicable diseases (NCDs) [8]. Ensuring access to sufficient testing capacity and

infection prevention and control measures for vulnerable groups such as refugees require a comprehensive approach to ensure safety of citizens and noncitizens.

Despite increasing evidence about COVID-19 and its spread globally, less is unknown about the direct and indirect effects of the virus in humanitarian and forced displacement settings. In this study, we report on the epidemiology of COVID-19 cases in Jordan's 2 largest refugee camps and evaluate the changes in routine health service utilization during the first year of the pandemic from April 1, 2020 to March 31, 2021.

## Methods

The study is reported as per the Strengthening the Report of Observational Studies in Epidemiology (STROBE) guideline (S1 Supporting Information).

### Study setting

Azraq and Zaatari refugee camps in Jordan hosted 37,932 and 79,034 Syrian refugees, respectively, as of March 2021. Children under 18 years constitute almost half of the population, while older persons (≥60 years) represent 4.2% [9].

### Data sources and study outcomes

The study used 2 sets of data by the United Nations High Commissioner for Refugees (UNHCR): (i) COVID-19 linelist; and (ii) routine health data from UNHCR's health information system (HIS).

COVID-19 linelists were compiled for each camp and included all laboratory confirmed COVID-19 cases between September 8, 2020 and April 2, 2021 (epidemiological week 13). Individual level information in each linelist included patient demographics (age, sex, nationality, and residence); test data (dates of sample collection, test, results; whether asymptomatic, and reason for testing); comorbidities or other underlying conditions; isolation and hospitalization (dates of admission and discharge, need for ventilation, intensive care, and oxygen); exposure risks (health worker status, travel outside of camp, visit to health facilities, and contact with confirmed case); disease outcome; and case contacts. Aggregated monthly testing data were provided by the UNHCR.

Jordan HIS reporting occurred weekly and covered the period from January 1, 2018 to April 2, 2021. Extracted data included number of outpatient consultations, antenatal care (ANC) visits, assisted deliveries, family planning consultations, new family planning consultations, number of administered "measles" vaccine doses, respiratory tract infections (RTIs; disaggregated by type: upper respiratory tract infection (URTI), lower respiratory tract infection (LRTI), and influenza-like illnesses (ILIs)), consultations for diabetes, consultations for injuries, and mortality. Complete definitions of indicators are provided in the Supporting information (Table A in S2 Supporting Information).

### Patient involvement

It was not possible to involve cases or communities living in the refugee camps as the analysis used existing anonymized and aggregated data.

### Statistical analysis

An analysis plan was developed during the study design phase (Methods in S2 Supporting Information). Descriptive statistics were calculated to describe COVID-19 case epidemiology. Comparisons of categorical variables were made with chi-squared tests or Fisher exact tests;

comparisons of continuous variables used *t* tests to detect differences in means between 2 categories (sex) and analysis of variance (ANOVA) tests to detect differences in means between multiple categories (age groups). Odds ratios for selected outcomes were calculated using generalized linear models (with binomial family link) and controlling for covariates: sex, age, and number of comorbidities by category. *p*-Values less than 0.05 were considered statistically significant. Comparisons between camps and with host country were explored. Analysis was conducted in R (Version 4.1.0) using RStudio v1.4.1106 [10].

We used interrupted time series to evaluate change in rates of consultations and other outcomes during the COVID-19 period. We used the model of the form

$$y_i = NB(y_i|\mu_i, \theta)$$

$$\log(\mu_i) = \alpha_i + \beta_1 COVID_i + \beta_2 Month\ since\ COVID_i + \beta_3(Centered\ Month_i)$$
$$+ \gamma_j Seasonal\ dummy_{ij} + \varepsilon_i + offset(\log(population_i)),$$

where

- *y* is outcome of interest, assumed to come from a negative binomial (NB) distribution with parameters $\mu_i$ and $\theta$;

- *Population$_i$* is number of people at risk or eligible to access relevant services at time *i*;

- *COVID$_i$* is binary variable (0 if month i is in pre–COVID-19 period and 1 if month *i* is in COVID-19 period);

- *Month since COVID$_i$* is time in months since beginning of COVID-19 (April 2020);

- *Centered Month$_i$* is month number, centered at beginning of the COVID-19 period, accounting for longer term trend; and

- *Seasonal dummy$_{ij}$* terms are 11 dummy terms to capture 12-month seasonality cycle.

For services where seasonality was unlikely to be a factor, we considered a model without seasonal dummy terms. For a few indicators where NB did not converge, we alternatively fit a model from a Gaussian family. For indicators where time lag was plausible, we assessed cross correlation between date and outcome indicator. Where a nonzero lag was observed, we considered a lagged model in sensitivity analysis. We assume AR1 model for residuals (Table B in S2 Supporting Information outlines model specifications for each indicator).

The model was fit as a generalized additive model using *mgcv* function in R. Because no smoothing terms were included, this is equivalent to using a generalized linear model, such as using the glm.nb() function in the *MASS* package. Point estimates and 95% confidence intervals (CIs; note that *mgcv* package uses Bayesian approach and, hence, results in a credible interval. However, as shown by Marra and Wood [32], in this context, we can treat the credible intervals in a frequentist manner, and the coverage probabilities will approximately hold. In the rest of the report, we will refer to these as confidence intervals for convenience) were obtained for $\beta_1$ and $\beta_2$ coefficient. A CI for $\beta_1$ that does not include 1 indicates an immediate change in outcome at beginning of COVID-19 period (i.e., a change in level or a step). An estimate below 1 indicates a decrease in outcome at beginning of COVID-19 period as compared to the counterfactual, and an estimate above 1 indicates a higher value of outcome than expected had COVID-19 not happened. A CI for $\beta_2$ that does not include 1 indicates a change in slope in evolution of outcome over time, accounting for long-term trend and seasonality, if applicable. A $\beta_2$ value greater than 1 indicates that values of outcomes were increasing faster in COVID-19 period or decreasing slower compared to pre–COVID-19 period.

For interrupted time series, potential outliers were defined as any values in the pre–COVID-19 period that were at least 3 standard deviations away from the pre–COVID-19 mean. These values were removed prior to analysis.

We report parameter estimates using incidence rate ratios (IRR) and related 95% CI; for visual comparison of evolution of outcome during COVID-19 period and expected evolution had COVID-19 not happened, we constructed the counterfactual. Counterfactual values are predicted by setting values of $COVID_i$ and $Month\ since\ COVID_i$ to 0, and forecasting the model for 12 months of COVID-19 period.

## Ethics approval

The study was determined as Non Human Subject Research by Johns Hopkins Bloomberg School of Public Health's Institutional Review Board (IRB number 19738). No in-country ethical approval was required as data were anonymized and aggregated.

## Results

### COVID-19 epidemiology

COVID-19 cases were first reported on September 8 and September 13, 2020 in Azraq and Zaatari, respectively. From beginning of the outbreak until April 2, 2021, 901 cases in Azraq and 1,715 in Zaatari were recorded among total midpoint populations of 37,462 and 78,281, respectively. Incidence rates (IRs) were lower in camps than neighboring governorates (Azraq IRR: 0.624, 95% CI: [0.584 to 0.666], p-value < 0.001 and Zaatari IRR: 0.598, 95% CI: [0.570, 0.629], p-value < 0.001) and Jordan (Azraq IRR: 0.403, 95% CI: [0.378 to 0.430], p-value < 0.001 and Zaatari IRR: 0.367, 95% CI: [0.350 to 0.385], p-value < 0.001) (Fig 1). Testing rates were higher than the national average in Azraq (IRR: 1.693, 95% CI: [1.671 to 1.715], p-value < 0.001) and lower than the national average in Zaatari (IRR: 0.678, 95% CI: [0.669 to 0.688], p-value < 0.001).

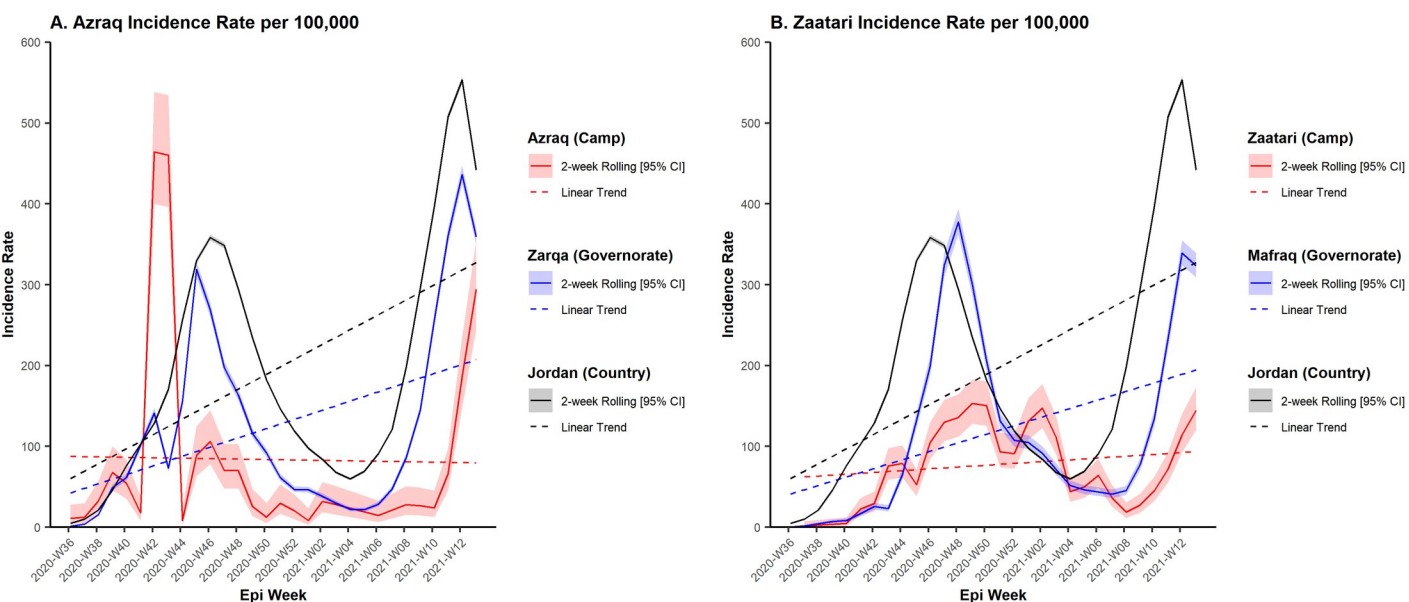

**Fig 1.** COVID-19 IR over time in Azraq camp (left) and in Zaatari camp (right) from epi wk 36 2020 –epi wk 12 2021 (2 week rolling average). CI, confidence interval; COVID-19, Coronavirus Disease 2019; IR, incidence rate.

**Table 1. Individual and population level characteristics of COVID-19 in Azraq and Zaatari refugee camps, Jordan (from first case in each camp–September 8, 2020 in Azraq and September 13, 2020 in Zaatari to April 2, 2021).**

| | Azraq | Zaatari |
|---|---|---|
| Individual-level characteristics–N (%) | | |
| Total number of cases | 901 | 1,715 |
| Sex distribution–female cases | 518 (57.5%) | 949 (55.3%) |
| Most affected age group | 30 to 39 years old 208 (23.1%) | 18 to 29 years old 360 (21.0%) |
| Proportion of cases with symptoms | 92 (10.2%) | 404 (23.6%) |
| Proportion of cases with comorbidities | | |
| 1 | 75 (8.3%) | 127 (7.4%) |
| 2 or more | 26 (2.9%) | 86 (5.0%) |
| Proportion of hospitalized cases with comorbidities | | |
| 1 | 17 (20.7%) | 27 (25.7%) |
| 2 or more | 14 (17.1%) | 33 (31.4%) |
| Disease outcomes | | |
| Hospitalization | 82 (9.1%) | 105 (6.1%) |
| Intensive care unit | NA | 18 (1.0% of total cases and 17.1% of hospitalized) |
| Ventilated | NA | 16 (0.9% of total cases and 15.2% of hospitalized) |
| Recovered | 892 (99.0%) | 1,691 (99.0%) |
| Case fatality rate | | |
| Crude | 9 (1.0%) | 17 (1.0%) |
| 60+ | 3 (33.3%) | 13 (76.4%) |
| Male cases | 6 (66.7%) | 9 (52.9%) |
| Comorbidities– 1 | 5 (55.6%) | 5 (29.4%) |
| Comorbidities– 2 or more | 4 (44.4%) | 9 (52.9%) |
| Population level parameters (March 1, 2020 to April 2, 2021) | | |
| IRs (per 100,000 persons) (CI) | | |
| Total camp population | 2,405 (2,255 to 2,565) | 2,191 (2,091 to 2,296) |
| 0 to 17 years | 1,416 (1,269 to 1,580) | 1,499 (1,388 to 1,618) |
| 18 to 59 years | 3,692 (3,398 to 4,010) | 2,968 (2,791 to 3,156) |
| 60+ | 6,882 (5,245 to 8,982) | 4,068 (3,306 to 4,997) |
| Testing rate (per 100,000 persons) | 98,187 | 39,367 |
| Test positivity rate | 2.4% | 5.4% |
| Time between sample collection and test results (mean number of days) | 4.5 | 2.4 |

CI, confidence interval; COVID-19, Coronavirus Disease 2019; IR, incidence rate.

Table 1 summarizes key descriptive results. Table 2 describes risk factors for selected outcomes. Older age (60+) is associated with higher odds of all negative outcomes: hospitalization (Azraq adjusted odds ratios (aORs): 2.30, 95% CI: [1.05 to 5.02], $p$-value = 0.04; Zaatari aOR: 2.33, 95% CI: [1.29 to 4.22], $p$-value = 0.01), admission to intensive care (Zaatari aOR: 11.28, 95% CI: [3.09 to 41.20], $p$-value < 0.001), ventilation (Zaatari aOR:7.77, 95% CI: [2.15 to 28.11], $p$-value < 0.001), and death (Zaatari aOR: 15.41, 95% CI: [4.03 to 59.02], $p$-value < 0.001). The exception to this was in Azraq where older age did not show a significantly higher odds of death. Intensive care and ventilation services were not available in Azraq. Presence of one comorbidity is associated with higher odds of hospitalization (Azraq aOR: 2.57, 95% CI: [1.31 to 5.05], $p$-value = 0.01; Zaatari aOR: 5.10, 95% CI: [2.95 to 8.83], $p$-value < 0.001), while 2 or more comorbidities with higher odds for all negative outcomes

**Table 2. aORs for disease outcomes, Azraq and Zaatari refugee camps, Jordan (from first case in each camp–September 8, 2020 in Azraq and September 13, 2020 in Zaatari to April 2, 2021).**

| | Azraq | | Zaatari | | | | |
|---|---|---|---|---|---|---|---|
| | Hospitalization | Death | Isolation | Hospitalization | Intensive care unit | Ventilation | Death |
| Male[1] | | | | | | | |
| aOR | 0.94 | 2.59 | 1.06 | 1.10 | 2.13 | 1.75 | 1.61 |
| 95% CI | 0.58 to 1.54 | 0.60 to 11.20 | 0.80 to 1.39 | 0.71 to 1.70 | 0.74 to 6.11 | 0.59 to 5.23 | 0.55 to 4.69 |
| p-value | 0.81 | 0.20 | 0.69 | 0.67 | 0.16 | 0.32 | 0.38 |
| Age 0 to 17[2] | | NA*: no deaths among 0 to 17 year olds | | | | NA*: no patient with outcome | NA*: no deaths among 0 to 17 year olds |
| aOR | 0.31 | | 1.06 | 0.15 | 0.78 | | |
| 95% CI | 0.15 to 0.64 | | 0.80 to 1.39 | 0.06 to 0.39 | 0.07 to 8.13 | | |
| p-value | <0.001 | | 0.69 | <0.001 | 0.83 | | |
| Age 60+[2] | | | | | | | |
| aOR | 2.30 | 0.67 | 0.28 | 2.33 | 11.28 | 7.77 | 15.41 |
| 95% CI | 1.05 to 5.02 | 0.14 to 3.16 | 0.17 to 0.47 | 1.29 to 4.22 | 3.09 to 41.20 | 2.15 to 28.11 | 4.03 to 59.02 |
| p-value | 0.04 | 0.61 | <0.001 | 0.01 | <0.001 | <0.001 | <0.001 |
| Comorbidities (1)[3] | | NA*: no deaths in patients without comorbidities | | | | | |
| aOR | 2.57 | | 0.48 | 5.10 | 5.91 | 2.42 | 4.69 |
| 95% CI | 1.31 to 5.05 | | 0.32 to 0.74 | 2.95 to 8.83 | 1.01 to 34.70 | 0.35 to 16.62 | 0.94 to 23.45 |
| p-value | 0.01 | | 0.01 | <0.001 | 0.08 | 0.37 | 0.06 |
| Comorbidities (2 +)[3] | | NA*: no deaths in patients without comorbidities | | | | | |
| aOR | 8.26 | | 0.32 | 9.11 | 18.45 | 15.35 | 7.64 |
| 95% CI | 3.21 to 21.25 | | 0.19 to 0.53 | 4.98 to 16.69 | 3.44 to 98.85 | 3.29 to 71.57 | 1.56 to 37.51 |
| p-value | <0.001 | | <0.001 | <0.001 | <0.001 | <0.001 | 0.01 |

*NA—not available, no patients with outcome.

[1]Ref: Female.

[2]Ref: Ages 18 to 59.

[3]Ref: Comorbidities (0).

aOR, adjusted odds ratio; CI, confidence interval.

(hospitalization (Azraq aOR: 8.26, 95% CI: [3.21 to 21.25], p-value = 0.01; Zaatari aOR: 9.11, 95% CI: [4.98 to 16.69], p-value < 0.001), admission to intensive care (Zaatari aOR: 18.45, 95% CI: [3.44 to 98.85], p-value < 0.001), ventilation (Zaatari aOR:15.35, 95% CI: [3.29 to 71.57], p-value < 0.001), and death (Zaatari aOR: 7.64, 95% CI: [1.56 to 37.51], p-value = 0.01). In Azraq, no deaths occurred in patients without comorbidities. Results about exposure risks and contacts are in S3 Supporting Information (Additional Results, Tables B to I).

## Changes in routine health services and other health outcomes during the COVID-19 pandemic

Overview of interrupted time series results are in Table 3 and Fig 2 below.

**Healthcare utilization rate.** Rates of total outpatient visits show marked reduction in both camps when the pandemic began (Fig 3). Azraq recorded an immediate decrease of 32%

**Table 3. Interrupted time series results: Impact of COVID-19 on routine health services and health outcomes in Azraq and Zaatari refugee camps, Jordan during the first year of the pandemic, January 1, 2018 to April 2, 2021.**

| | Azraq | | | | Zaatari | | | |
|---|---|---|---|---|---|---|---|---|
| | IRR immediate change [95% CI] | p-Value | IRR change in trend [95% CI] | p-Value | IRR immediate change [95% CI] | p-Value | IRR change in trend [95% CI] | p-Value |
| Health utilization rate | 0.680 [0.549 to 0.843] | <0.001 | 1.028 [1.003 to 1.053] | 0.030 | 0.758 [0.577 to 0.995] | 0.046 | 0.978 [1.011 to 1.011] | 0.188 |
| Mortality | 0.476 [0.235 to 0.965] | 0.039 | 1.078 [0.995 to 1.169] | 0.066 | 1.052 [0.731 to 1.512] | 0.786 | 1.016 [0.975 to 1.059] | 0.451 |
| RTIs | | | | | | | | |
| LRTIs | 1.159 [0.877 to 1.532] | 0.298 | 0.915 [0.885 to 0.945] | <0.001 | 1.597 [0.977 to 2.612] | 0.062 | 0.783 [0.740 to 0.829] | <0.001 |
| URTIs | 0.693 [0.561 to 0.855] | 0.001 | 0.971 [0.947 to 0.995] | 0.020 | 0.604 [0.446 to 0.818] | 0.001 | 0.972 [0.937 to 1.007] | 0.117 |
| ILIs | 0.764 [0.354 to 1.649] | 0.500 | 0.891 [0.801 to 0.992] | 0.047 | 0.673 [0.383 to 1.183] | 0.182 | 0.892 [0.808 to 0.984] | 0.032 |
| All RTIs | 0.749 [0.596 to 0.940] | 0.013 | 0.949 [0.924 to 0.975] | <0.001 | 0.625 [0.461 to 0.849] | 0.003 | 0.936 [0.904 to 0.970] | <0.001 |
| NCDs | | | | | | | | |
| Diabetes | 1.166 [0.766 to 1.774] | 0.474 | 0.996 [0.926 to 1.072] | 0.919 | 0.992 [0.723 to 1.360] | 0.959 | 0.996 [0.958 to 1.034] | 0.821 |
| Injury | 0.429 [0.266 to 0.693] | 0.001 | 0.941 [0.889 to 0.996] | 0.036 | 1.164 [0.699 to 1.941] | 0.559 | 1.037 [0.976 to 1.102] | 0.239 |
| Reproductive health | | | | | | | | |
| Family planning–old and new consultations | 0.526 [0.376 to 0.736] | <0.001 | 0.977 [0.938 to 1.018] | 0.266 | 0.524 [0.312 to 0.878] | 0.014 | 1.144 [1.075 to 1.218] | <0.001 |
| New family planning only | 0.532 [0.329 to 0.861] | 0.010 | 1.071 [1.011 to 1.135] | 0.020 | 0.595 [0.305 to 1.162] | 0.128 | 1.073 [0.990 to 1.164] | 0.088 |
| Maternal and child health | | | | | | | | |
| Ante-natal care 1 coverage | 0.793 [0.558 to 1.127] | 0.196 | 1.027 [0.985 to 1.072] | 0.213 | 0.659 [0.336 to 1.294] | 0.226 | 1.118 [1.031 to 1.213] | 0.007 |
| Live births coverage | 1.032 [0.822 to 1.295] | 0.786 | 0.997 [0.970 to 1.025] | 0.832 | 1.090 [0.875 to 1.358] | 0.442 | 1.001 [0.975 to 1.028] | 0.929 |
| Measles vaccination coverage | 0.717 [0.506 to 1.015] | 0.061 | 1.048 [1.006 to 1.093] | 0.025 | 0.752 [0.556 to 1.015] | 0.063 | 1.014 [0.978 to 1.052] | 0.450 |

Cells highlighted in gray indicate a statistically significant result (CI does not include 1).

CI, confidence interval; COVID-19, Coronavirus Disease 2019; ILI, influenza-like illness; IRR, incidence rate ratio; LRTI, lower respiratory tract infection; NCD, noncommunicable disease; RTI, respiratory tract infection; URTI, upper respiratory tract infection.

(IRR: 0.680, 95% CI [0.549 to 0.843], p-value < 0.001) and Zaatari of 24% (IRR: 0.758, 95% CI [0.577 to 0.995], p-value = 0.046). Both camps show qualitatively similar results together with good model fits. While before COVID-19, health utilization rate had a long-term decreasing trend, when the pandemic began, there was an immediate drop that was not explained by this longer trend, accounting for long-term trend and seasonality. Estimates for change in slope during the COVID-19 period are very close to 1, indicating that the slope of the utilization rate remained relatively stable during COVID-19 period compared to pre–COVID-19 period. In Azraq, the CIs for change in slope do not include 1 (IRR: 1.028, 95% CI [1.003 to 1.053], p = 0.030).

**RTIs.** RTIs evolved similarly in Azraq and Zaatari (Fig 4). In both camps, all RTIs show an immediate decrease when the pandemic began: in Azraq by 25% (IRR: 0.749, 95% CI: [0.596 to 0.940]; p-value = 0.013) and in Zaatari by 37% (IRR: 0.625, 95% CI: [0.461 to 0.849]; p-value = 0.003). Only CIs for URTI do not include 1 in both camps: In Azraq, we see a

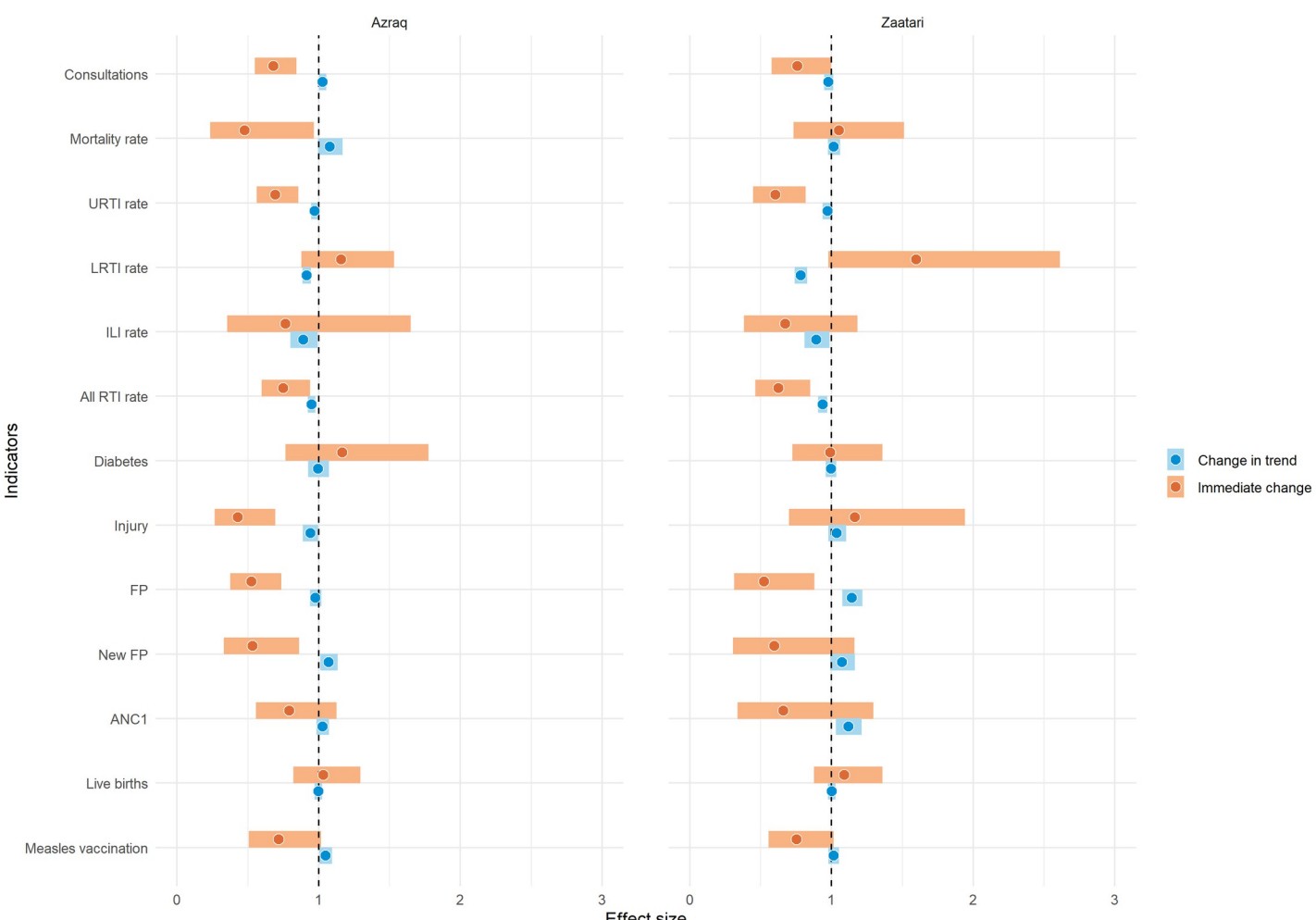

**Fig 2. Overview of interrupted time series results on health services and health outcomes in Azraq and Zaatari refugee camps, Jordan during the first year of the COVID-19 pandemic, January 1, 2018 to April 2, 2021.** Note: The dot indicates the IRR estimate, and the bar the CIs. CIs encompassing 1 (dotted vertical line) indicate results that are not statistically significant. ANC1, first antenatal care visit; CI, confidence interval; COVID-19, Coronavirus Disease 2019; FP, family planning; ILI, influenza-like illness; IRR, incidence rate ratio; LRTI, lower respiratory tract infection; RTI, respiratory tract infection; URTI, upper respiratory tract infection.

reduction by 31% (IRR: 0.693, 95% CI: [0.561 to 0.855]; *p*-value = 0.001) and in Zaatari, by 40% (IRR: 0.604, 95% CI: [0.446 to 0.818]; *p*-value = 0.001). Change in slope over time is negative for all types of RTIs in Azraq, with decreases ranging from 5% for all RTIs to 11% for ILIs. Change in slope is also negative and significant in Zaatari for all but URTI. Decreases range between 22% for LRTI and 7% for RTI. LRTI show an increase in both camps, but results are not sustained over time, and results are not statistically significant. ILIs show an immediate decrease in both camps (not statistically significant) and a significant decreasing slope.

Model fit is satisfactory for all RTIs except for ILIs, for which a Gaussian model is used. CIs are narrow for RTIs and URTIs in both camps, but less so for LRTIs (especially in Zaatari) and ILIs (especially in Azraq). Sensitivity analysis results are robust (Table A in S3 Supporting Information, Additional Results).

**NCDs.** Number of diabetes consultations are unstable over the study period, which is reflected in the large CI (Fig A in S3 Supporting Information, Additional Results) and results that are not statistically significant for any of the coefficients. In Azraq, diabetes consultation

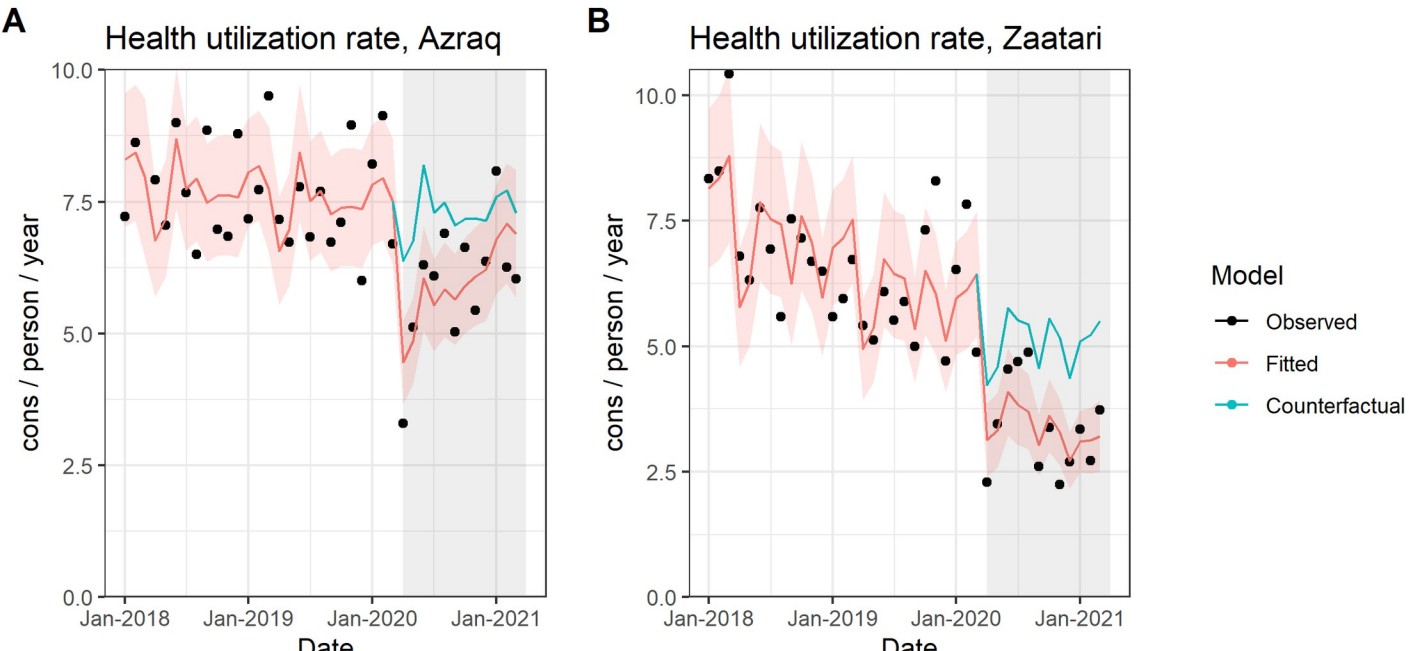

**Fig 3.** Time series of health utilization rate (expressed as consultations/per person/per year) in Azraq (**A**) and Zaatar (**B**) camps Jordan, January 1, 2018 to April 2, 2021.

rate increases when the pandemic began, but results are not statistically significant. In Zaatari, we observe a slight immediate decrease when the pandemic began, and a relatively stable trend in diabetes consultations in line with the preexisting trend. Similarly, the change in slope does not differ from the pre–COVID-19 period.

Consultations for injuries show 2 different patterns. In Azraq, an increasing trend in consultations for injuries is observed since 2018, which is interrupted when the pandemic began. Consultations for injuries decrease by 57% (IRR: 0.429, 95% CI: [0.266 to 0.693], *p*-value = 0.001), with a decreasing change in slope over time (IRR: 0.941, 95% CI: [0.889, 0.996], *p*-value = 0.036). In Zaatari, consultations for injuries have declined since 2018 and have an immediate increase of 16% (not statistically significant) at the beginning of the COVID-19 period. The slope indicates a slight increase from pre–COVID-19 (IRR: 1.037), but the change is not statistically significant.

**Maternal and child health.** Values of ANC1 coverage are high and unstable during entire study period, reaching more than 250% in early 2019 in Azraq and more than 400% in Zaatari in January 2021 (Fig 5). CIs are wide, and model fit is mediocre. There is an immediate drop in both Azraq and Zaatari (Fig 5A and 5B), but results are not statistically significant. Change in slope over time is slightly greater than 1 in Azraq and shows an increase by 12% in Zaatari (IRR: 1.118, 95% CI: [1.031 to 1.213]).

Extensive variability is recorded in live birth coverage in both camps over study period (Fig 5C and 5D). This negatively affects model fit and is reflected in wide CIs. While an immediate increase is observed in both camps, results are not statistically significant. Changes in slopes over time do not differ much from pre–COVID-19 period.

An immediate drop in measles vaccine doses distributed is observed in both camps (Fig 5E and 5F): Azraq reports a 28% decrease (IRR: 0.717, 95% CI: [0.506 to 1.015], *p*-value = 0.061) and Zaatari a 25% decrease (IRR: 0.752, 95% CI: [0.556 to 1.015], *p*-value = 0.063); both CIs

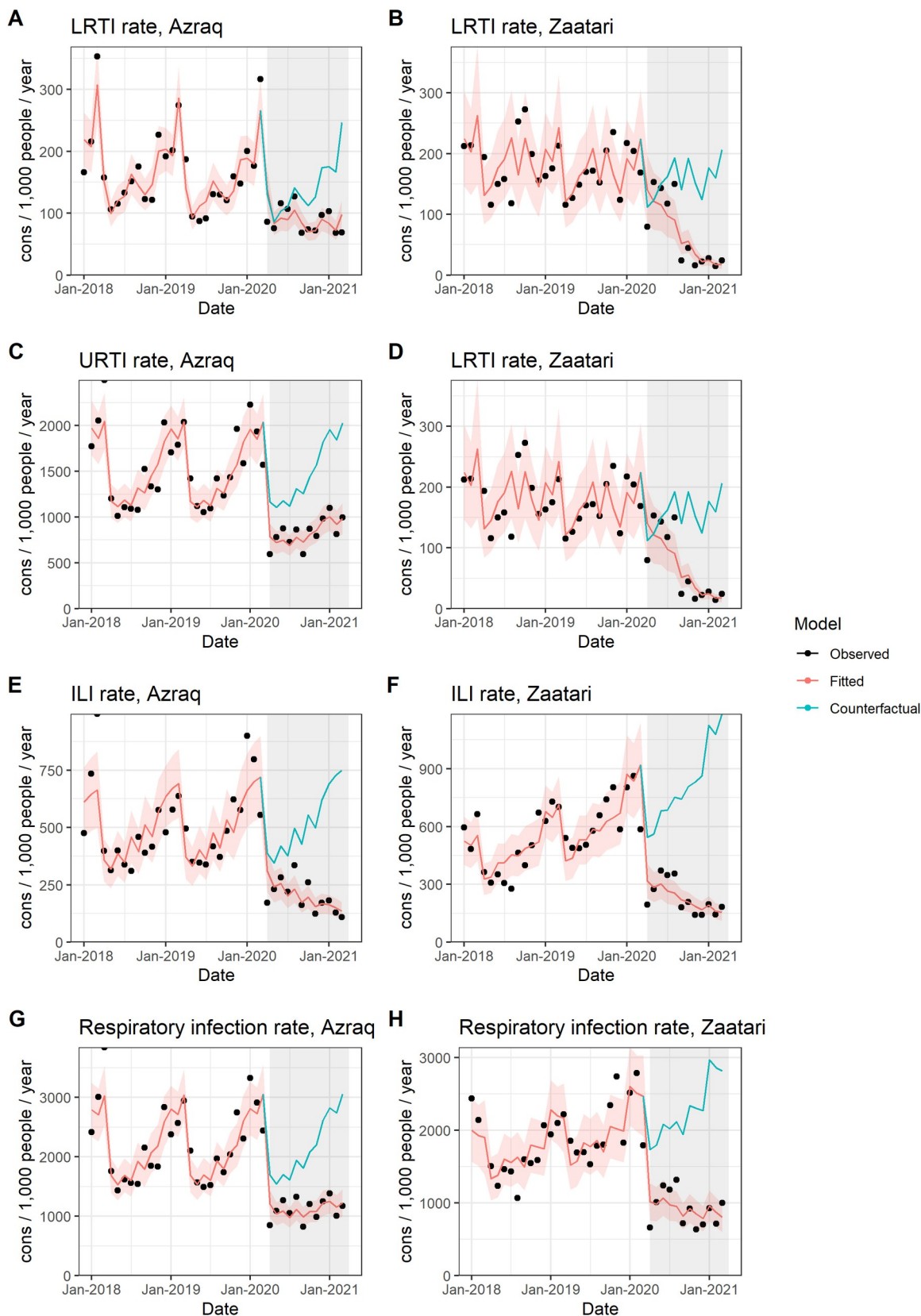

**Fig 4.** Interrupted time series results for RTI, Azraq and Zaatari camps, Jordan, January 1, 2018 to April 2, 2021: LRTIs (**A, B**); URTI (**C, D**); ILI (**E, F**); and All RTIs (**G, H**). ILI, influenza-like illness; LRTI, lower respiratory tract infection; RTI, respiratory tract infection; URTI, upper respiratory tract infection.

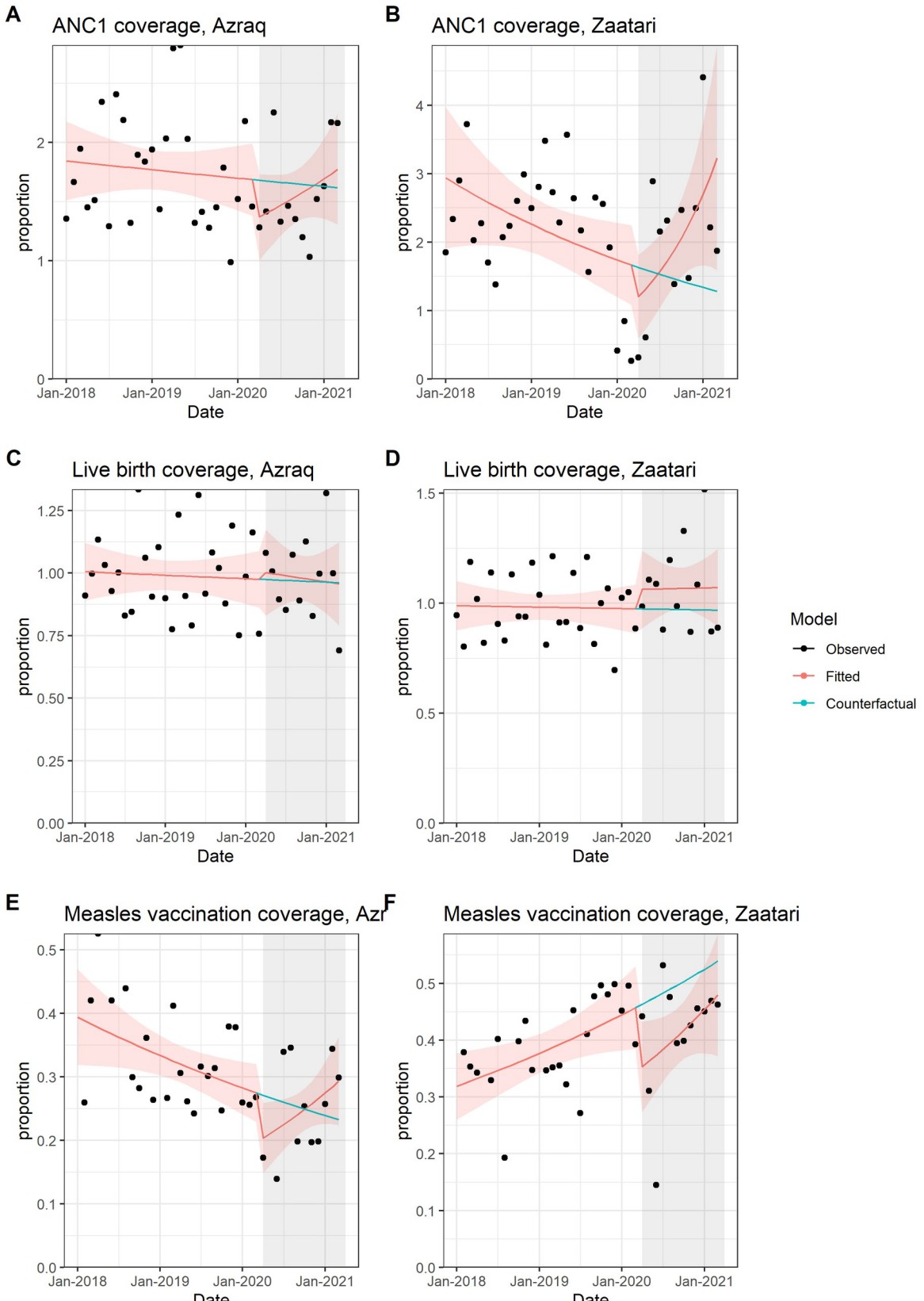

**Fig 5.** Interrupted time series for maternal and child health indicators, Azraq and Zaatari camps, Jordan (January 1, 2018 to April 2, 2021): ANC1 **(A, B)**; live birth coverage **(C, D)**; and measle vaccination coverage **(E, F)**. ANC1, first antenatal care visit.

include 1. Change in slope over time is greater than 1 in Azraq (IRR: 1.048, 95% CI: [1.006 to 1.093], $p$-value = 0.025) and very close to 1 in Zaatari (IRR: 1.014, 95% CI: [0.978 to 1.052], $p$-value = 0.450).

**Reproductive health.** Family planning indicators report an immediate drop when the pandemic began in both camps (Fig B in S3 Supporting Information, Additional Results). New and repeat family planning consultations decrease by 47% in Azraq (IRR: 0.526, 95% CI: [0.376 to 0.736], $p$-value < 0.001) and 48% in Zaatari (IRR: 0.524, 95% CI: [0.312 to 0.878], $p$-value = 0.014). New consultations decrease by 47% in Azraq (IRR: 0.532, 95% CI: [0.329 to 0.861], $p$-value = 0.010) and by 40% in Zaatari (IRR: 0.595, 95% CI: [0.305 to 1.162], $p$-value = 0.128). Estimations for Zaatari are less stable than for Azraq due to high variability in values, reflected in wide CIs, poor model fit, and a nonstatistically significant result for new consultations. Change in slope over time is positive for the new family planning indicators in each camp; however, it differs from pre–COVID-19 trend for new consultations only in Azraq (IRR: 1.071, 95% CI: [1.011 to 1.135], $p$-value = 0.020). For Zaatari, there is an increase change of 7% per month since beginning of COVID-19 period, but the CI is wide, and results are not statistically significant (IRR: 1.073, 95% CI: [0.990 to 1.164]).

**Mortality.** Absolute number of deaths in both camps over the entire study period is low: 5.9 registered deaths per month in Azraq on average (1.93 deaths/1,000 people/year) and 12 in Zaatari (1.85 deaths/1,000 people/year). Consequently, model fit is problematic for Azraq (Box–Ljung $p$-value = 0.0205), and CIs are large. While in Zaatari, coefficients are not statistically significant at 0.05 level, in Azraq, there is a marked immediate drop (IRR: 0.476, 95% CI: [0.235 to 0.965], $p$-value = 0.039), followed by a a slight increase in slope (IRR: 1.078, 95% CI: [0.995 to 1.169], $p$ = 0.066).

## Discussion

COVID-19 cases were first reported in Azraq and Zaatari 6 months after the first reported case in Jordan. Observed viral transmission, measured by IRs, was lower in the refugee camps compared to their respective governorates and Jordan during the same time period (Table 4); in Zarqa (where Azraq is located), the IR was 3,856/100,000 persons, and in Mafraq (where Zaatari is located), it was 3,660/100,000 persons. Both camps' IRs were lower than the national level in Jordan (5,974/100,000 persons) for the same time period [11]. IRs in Azraq and Zaatari

**Table 4. Comparison of incidence and testing rates between refugee camps, governorates, and national level, Jordan, March 1, 2020 to April 2, 2021.**

|  | COVID-19 cases | Population | Incidence/100,000 persons | Testing rate/100,000 persons |
|---|---|---|---|---|
| **Jordan** | 609,453[1] | 10,203,140[4] | 5,974.1 | 58,003 |
| **Jordan** | 594,094[2] | 10,203,140[4] | 5,822.7 | 58,003 |
| **Mafraq governorate** | 20,130[2] | 549,948[5] | 3,660.3 | NA |
| **Zarqa governorate** | 52,632[2] | 1,364,878[5] | 3,856.2 | NA |
| **Azraq camp** | 901[3] | 37,462[6] | 2,405.1 | 98,187 |
| **Zaatari camp** | 1,715[3] | 78,281[6] | 2,190.8 | 39,367 |

[1]Johns Hopkins COVID Research Center [12].

[2]COVID-19 data from MoH Jordan Daily Updates [13].

[3]UNHCR Linelist.

[4]World Bank [14] (accessed 2021 Oct 12).

[5]Government of Jordan [15] (accessed 2021 Oct 12).

[6]UNHCR HIS data (midpoint population).

COVID-19, Coronavirus Disease 2019; HIS, health information system; UNHCR, United Nations High Commissioner for Refugees.

followed similar, but delayed, trends in their respective governorates and in Jordan (Fig 1). Spikes in Zaatari followed increases in Mafraq. Azraq also followed increases in Zarqa, except for a spike in epi week 43 that is likely an artifact due to delayed testing of previous weeks. Consequently, Azraq and Zaatari were likely affected by outside transmission rather than driving the epidemic in the governorate.

Lower IRs in Zaatari could be partially explained by lower testing capacity than the national level (testing rate was 58,003/100,000 persons in Jordan compared to 39,367 in Zaatari during the study period). However, the testing rate in Azraq was twice the national level (98,187/ 100,000 persons), suggesting a lower infection rate. Testing (as well as hospitalization and vaccination) was free for refugees as they were included in the National Preparedness and Response COVID-19 plan, whose whole-of-society approach meant both Jordanians and non-Jordanians living in host communities and in camps had the same access to health services [16]. Furthermore, a positive test result did not have any implications in terms of status or access to service or livelihood supports, besides the need to isolate (as with nationals). Barriers to testing were, therefore, mainly on the supply side (i.e., testing availability). The Government of Jordan's effort to integrate refugees into the national response plan is an important step toward universal health coverage and an important path toward ending the pandemic.

These results support the assertion that refugee populations living in the 2 largest camps in Jordan did not represent a threat of spreading Severe Acute Respiratory Syndrome Coronavirus 2 (SARS-CoV-2) among the general population, as has been claimed in other countries hosting refugees [17,18] While this paper contributes to the evidence about true infection risk from refugees and the lack of association between migration and spread of diseases, refugees were actually not stigmatized or discriminated in Jordan. On the contrary, refugees were included in the national COVID-19 response plan since the beginning and had same access to testing, treatment, and then vaccine as nationals.

Strict measures such as movement restrictions within and outside the camps, bans on gatherings, curfews, closure of shops during weekends, wearing masks, and maintaining physical distance were implemented in the camps since early March 2020, well before the first cases were recorded (see S4 Supporting Information NPIs for a detailed description of the measures introduced in camps). These nonpharmaceutical interventions (NPIs) were likely important factors in successfully delaying the introduction of COVID-19 among refugees in the camps and reducing its transmission. Such measures may have been easier to implement in enclosed and regimented settings and, therefore, could have been more effective within camps than among the general population and out-of-camp refugees in Jordan. More community behavioral data are needed to confirm this hypothesis. Furthermore, this is no indication that NPIs are more easily accepted or acceptable in camps, nor that camp-like settings are a desirable solution to prevent the spread of diseases. Durable and integrated solutions that strive for better living standards without mobility limitations remains UNHCR's long-term goals for refugees.

Characteristics of cases and risk factors for negative disease outcomes were consistent with increasing evidence about the COVID-19 disease [19]. In both camps, older age was a risk factor for hospitalization, as was admission to intensive care unit, ventilation, and death for Zaatari. Comorbidities, especially multiple comorbidities, were associated with higher odds for all adverse outcomes in both camps.

Evidence of indirect effects due to COVID-19 in refugee camps is limited with no published studies to our knowledge attempting to quantitatively estimate the effect of the pandemic on health service provision. Our analysis showed that changes in health services during the pandemic COVID-19 varied across services, but were similar in most respects for both camps. Overall healthcare utilization reported an immediate drop in both camps when the pandemic began, likely due to changes in health seeking behavior, since services were still functioning.

However, absolute numbers of visits remained high pre- and during the pandemic when compared with Sphere standard of 2 to 4 consultations/person/year [20]. In Azraq, healthcare utilization decreased from 7.7 consultations/person/year in pre–COVID-19 period to 6.2 during COVID-19. In Zaatari, it decreased from 6.5 to 3.4. However, this cannot be entirely attributable to COVID-19 only, as there was a declining trend since 2018. Reductions in health utilization rates due to COVID-19 have been recorded in other low- and middle-income countries in nonrefugee settings such as Kenya and Uganda [21,22].

Consultations for all types of RTIs declined in both camps. This decrease could relate to positive externalities of public health and social measures implemented to limit COVID-19 spread, which may have reduced transmission of non–COVID-19 RTIs as seen in other countries [23,24]. It could also indicate changes in health seeking behavior that led to reduced care for RTIs, possibly due to fear of being tested, isolated, and quarantined due to COVID-19. LRTIs consultations increased in both camps (results are not statistically significant), but data do not allow us to correlate this with COVID-19–related deaths. Decreases in consultations for injuries was observed, mainly in Azraq. Movement restrictions and reduced work, sport, or other physical activities could have caused a reduction in accidents, and, consequently, injuries, which has been observed in other settings [25,26]. This was also observed in Zaatari, where the trend was already decreasing from mid-2018.

Analyzing how maternal health services changed during COVID-19 pandemic was complicated due to variability in the pre–COVID-19 period. Values for ANC1 are often overestimated as women may seek care in multiple facilities. While a reduction in ANC1 was observed when the pandemic began, the historic trend is difficult to disentangle from COVID-19 due to high variability. Similarly, live births attended by health personnel did not seem to be affected by COVID-19. The impacts of COVID-19 on maternal health services have been diverse in non refugee settings ranging from no changes reported in the Democratic Republic of Congo except for urban centers [27,28], to positive changes in Kenya [29], with mixed effects in Uganda [21,22]. UNHCR and its partners attempted to maintain maternal health services by switching to a hybrid delivery model including telemedicine to reduce exposure for pregnant women; this could have had a positive effect on maintaining services. Immediate reductions were observed in both camps for family planning services for new consultations and existing clients. This may reflect a reluctance to visit health facilities that resolved over time and movement limitations for nonurgent care. However, since drug prescriptions were extended from 1 to 3 months from April 2020, such a drop in consultations may have limited disruption in contraception utilization. Interviews of family planning users could shed light on this aspect, but were beyond the scope of this study.

Other preventive measures such as measles immunization appeared to be negatively affected. There was an immediate drop when the pandemic began (however, results are significant at 0.10 level only), which may be explained by various factors including the closure of all vaccination clinics for 2 weeks in March 2020, movement restriction measures, and delays in seeking vaccination and its reporting. Fortunately, measles vaccination services showed a positive trend over time during the COVID-19 period, indicating that disruptions in vaccination services were temporary.

Ensuring access to continued NCD services a priority for UNHCR. Interpreting results related to diabetes consultations was challenging due to data variability before the COVID-19 period. While Azraq reported an increase and Zaatari a decrease in consultations, both results are not statistically significant. UNHCR quickly switched to 3-month prescriptions, which likely helped to maintain a high rate of treatment adherence without visiting health facilities, as did the switch to telemedicine for mental health, diabetes, and other NCD consultations.

Conducting interrupted time series analyses in volatile settings such refugee camps can be challenging as it may be difficult to meet some analytical assumptions and to mitigate threats to validity. First, treating the COVID-19 period as a single uniform period with a clear starting point may not capture the dynamics of this time, as different NPIs were introduced or lifted, transmission patterns varied, and attitudes and behavior changed. Second, other events that affected outcomes may have occurred both in pre–COVID-19 period and in the COVID-19 period. Factors such as policy changes, arrival of new populations with different health profiles, and changes in funding for service provision made it difficult to establish a pre–COVID-19 comparison. For example, mortality was unexpectedly elevated in the first half of 2018 in both camps. ANC1, deliveries, and diabetes consultations showed important heterogeneity or erratic patterns, which reduced the model fit and limited the capacity of the analysis to identify changes. Third, COVID-19 is one of the many factors affecting displacement settings, which makes it difficult to identify a "normal" time with which to compare. Finally, seasonality may have been a time-varying confounder that varied over years, and the autocorrelation structure of order 1 used in analysis may not adequately capture autocorrelation. Other confounders or effect modifiers may have been important, but are not considered in the model.

Unlike the out-of-camp refugees in Jordan, refugees in Azraq and Zaatari had access to functioning and free health services before and during the COVID-19 pandemic. Although refugees could not leave the camps for much of the pandemic due to movement restrictions, they still had access to food and voucher distribution systems that facilitated refugees' access to food, thus limiting economic hardship compared to out-of camp refugees and Jordanians. While more Syrian refugees than Jordanians were already living below the poverty line pre–COVID-19, the poverty gap increased less for registered Syrian refugees, particularly those living in the camps who are more reliant on UN and nongovernmental organization support than the labor market [30]. Althought the Government of Jordan introduced measures to continue the provision of essential health services for both Jordanians and out-of-camp refugees, there are no available data are to analyze its effects.

The findings from Jordan's camps cannot be generalized to other more precarious forced displacement situations. For example, infection rates among refugees and asylum seekers in crowded reception facilities in Greece were higher than among the general Greek population in 2020 [31]. Living conditions in such facilities, however, were particularly poor with limited access to water, sanitation, and healthcare facilities with no space to isolate. Ensuring acceptable living standards and equitable access to healthcare for refugees, as well as a functioning and inclusive surveillance, testing, treatment, and vaccination system is paramount to reduce the risk of infection among refugees and general population.

In conclusion, these insights into Jordan's refugee camps during the first year of the COVID-19 pandemic set the stage for follow-up research to investigate how infection susceptibility evolved over time, as well as which mitigation strategies were more successful and accepted. The pandemic has both exacerbated existing inequalities and demonstrated that until all populations are included in national response plans, the world remains vulnerable to the current and the next pandemic.

## Supporting information

**S1 Supporting Information. STROBE Checklist.** STROBE, Strengthening the Report of Observational Studies in Epidemiology.
(DOCX)

**S2 Supporting Information. Methods. Table A:** Definition of outcome indicators included in the interrupted time series analysis. **Table B:** Model specification for interrupted time series

analysis.
(DOCX)

**S3 Supporting Information. Additional results. Table A:** Interrupted times series results from alternative estimation models for RTIs, Azraq and Zaatari camps, Jordan. **Fig A:** Interrupted time series for diabetes consultations, Azraq and Zaatari camps, 2018 to 2021. **Fig B:** Interrupted time series of reproductive health indicators, Azraq and Zaatari camps, Jordan, January 1, 2018 to April 2, 2021: All family planning consultations (panels A and B); new family planning consultations (panels C and D). **Table B:** Proportion of COVID-19 cases by exposure type (all cases and by age groups), Azraq camp. **Table C:** Proportion of cases by setting of contact with other COVID-19 cases, by sex, Azraq camp. **Table D:** Proportion of COVID-19 cases by exposure type (all cases and by age groups), Zaatari camp. **Table E:** Proportion of cases by setting of contact with other COVID-19 cases, by sex, Zaatari camp. **Table F:** Average number of contacts per COVID-19 case, by sex and age groups, Azraq camp. **Table G:** Average number of contacts per COVID-19 case, by sex and age groups, Zaatari camp. **Table H:** Proportion of contacts followed by sex and age group of the case, Azraq camp. **Table I:** Proportion of contacts followed by sex and age group of the case, Zaatari camp. COVID-19, Coronavirus Disease 2019; RTI, respiratory tract infection.
(DOCX)

**S4 Supporting Information. NPIs.** NPI, nonpharmaceutical intervention.
(DOCX)

## Acknowledgments

The authors would like to thank Ann Burton (UNHCR) and Michael Woodman (UNHCR) for their support during the study.

## Author Contributions

**Conceptualization:** Chiara Altare, Paul B. Spiegel.

**Data curation:** Heba Hayek.

**Formal analysis:** Chiara Altare, Natalya Kostandova, Jennifer OKeeffe.

**Funding acquisition:** Paul B. Spiegel.

**Methodology:** Chiara Altare, Natalya Kostandova, Paul B. Spiegel.

**Project administration:** Chiara Altare.

**Validation:** Paul B. Spiegel.

**Visualization:** Chiara Altare, Natalya Kostandova, Jennifer OKeeffe.

**Writing – original draft:** Chiara Altare, Paul B. Spiegel.

**Writing – review & editing:** Chiara Altare, Natalya Kostandova, Jennifer OKeeffe, Heba Hayek, Muhammad Fawad, Adam Musa Khalifa, Paul B. Spiegel.

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
