## [Editor Report · Decision Letter 0]

1 Feb 2022

Dear Dr Spiegel, 

Thank you for submitting your manuscript entitled "COVID-19 pandemic in Azraq and Zaatari refugee camps in Jordan: are refugees in camps at reduced risk?" for consideration by PLOS Medicine.

Your manuscript has now been evaluated by the PLOS Medicine editorial staff as well as by an academic editor with relevant expertise and I am writing to let you know that we would like to send your submission out for external peer review.

Please re-submit your manuscript within two working days, i.e. by Feb 03 2022 11:59PM.

Kind regards,

Beryne Odeny

PLOS Medicine

---

## [Decision Letter · Decision Letter 1]

3 Mar 2022

Dear Dr. Spiegel,

Thank you very much for submitting your manuscript "COVID-19 pandemic in Azraq and Zaatari refugee camps in Jordan: are refugees in camps at reduced risk?" (PMEDICINE-D-22-00341R1) for consideration at PLOS Medicine. 

Your paper was evaluated by a senior editor and discussed among all the editors here. It was also sent to independent reviewers, including a statistical reviewer. The reviews are appended at the bottom of this email and any accompanying reviewer attachments can be seen via the link below:

[LINK]

In light of these reviews, I am afraid that we will not be able to accept the manuscript for publication in the journal in its current form, but we would like to consider a revised version that addresses the reviewers' and editors' comments. Obviously we cannot make any decision about publication until we have seen the revised manuscript and your response, and we plan to seek re-review by one or more of the reviewers. 

We expect to receive your revised manuscript by Mar 24 2022 11:59PM. Please email us (plosmedicine@plos.org) if you have any questions or concerns.

We look forward to receiving your revised manuscript. 

Sincerely,

Beryne Odeny, 

PLOS Medicine

plosmedicine.org

1) Please revise your title according to PLOS Medicine's style. Your title must be nondeclarative and not a question. It should begin with main concept if possible. Please place the study design in the subtitle (i.e., after a colon), e.g., a retrospective cohort study. Include setting in the title. For example, “COVID-19 epidemiology and health service utilization among refugees in Azraq and Zaatari refugee camps in Jordan: a retrospective cohort study”

2) Is there a chance you can obtain more recent data from this setting especially after vaccine roll out?

3) We note that you conducted research or obtained samples in a foreign country. Did you receive local IRB or ethical approval? Please check the relevant national regulations and laws applying to foreign researchers and state whether you obtained the required permits and approvals. Please address this in your ethics statement in both the manuscript and submission information.

4) The Data Availability Statement (DAS) requires revision. For each data source used in your study, if the data are not freely available, please describe briefly the ethical, legal, or contractual restriction that prevents you from sharing it. Please also include an appropriate contact (web or email address) for inquiries (this cannot be a study author).

5) Abstract:

a) Please ensure that all numbers presented in the abstract are present and identical to numbers presented in the main manuscript text.

b) Please include the population size

c) Please include the actual amounts and/or absolute risk(s) of relevant outcomes, not just coefficients.

d) Please quantify the main results (please present both 95% CIs and p values).

e) In the last sentence of the Abstract Methods and Findings section, please describe the main limitation(s) of the study's methodology.

f) Please address the study implications without overreaching what can be concluded from the data.

7) Please avoid assertions of primacy (“This is the first study to ..."). Instead, state “to our knowledge” or similar.

8) Please remove the last sentence of your introduction (“We adjust our analyses …”) and move it the Methods section 

9) Did your study have a prospective protocol or analysis plan? Please state this (either way) early in the Methods section. 

10) Please ensure that the study is reported according to the STROBE guideline, and include the completed STROBE checklist as Supporting Information. Please add the following statement, or similar, to the Methods: "This study is reported as per the Strengthening the Reporting of Observational Studies in Epidemiology (STROBE) guideline (S1 Checklist)."

11) Please provide p values in addition to 95% CIs in the main text and tables

12) Your study is observational and therefore causality cannot be inferred. Throughout the text, please remove language that implies causality, such as “impact” and “effects.” Refer to associations instead.

13) Please define the abbreviations in Tables and Figures e.g, OR, IRR, CI, cons, FP, URTI, LRTI, ILI, ANC1

14) In figure 2 and parts of figure 3, please show the axis beginning at zero for both graphs. If this is not possible, please show a break in the axis.

15) Please do not report P<0.0001; report as P < 0.001.

16) Please remove the “Role of the funding source” statement in the methods section. This information is captured in the metadata obtained in the submission form

17) Please replace the term “compliance” with “adherence” where it is used to refer to treatment adherence.

18) In the conclusion of the discussion, please present implications and next steps for research, clinical practice, and/or public policy; one-paragraph conclusion.

19) Please remove the “Competing Interests”, “Data availability,” and “Funding” statements at the end of the main text. This information is captured in the metadata obtained in the submission form.

20) Please move “Ethics Approval” to the Methods section.

21) References: 

a) Please select the PLOS Medicine reference style in your citation manager. In-text reference call outs should be presented as follows noting the absence of spaces within the square brackets, e.g., "... services [1,2]."

b) References should have no less and no more than six names before et al. For those with more than six names, please ensure that et al., is inserted after six names

c) Please ensure that journal name abbreviations consistently match those found in the National Center for Biotechnology Information (NCBI) databases. https://journals.plos.org/plosmedicine/s/submission-guidelines#loc-references. 

Comments from the reviewers:

Reviewer #1: Dear Authors,

Thanks a lot for this very interesting and informative article. It contains such an important information in the care of refugees in the humanitarian setting and should be published.

I have three comments (and inquiries).

First is the covid-19 diagnostic process and treatment. I assume Syrian refugees in both camps have full and free access to covid-19 diagnosis (PCR) and treatment (hospitalization when needed). I suggest elaborating this, if not yet, in the manuscript. When such access is not free nor easy, people may behave differently: i.e., may not seek PCR tests. I am raising this to exclude the possibility that lower incidence rate is not the product of limited access to PCR & treatment. For Jordanians, it's free in both PCR tests and hospitalizations at government facilities. Is this the same for Syrian refugees in both camps?

If PCR & hospitalization access is free for Syrian refugees, I would appreciate if you could mention this, with possible appraisal, in the manuscript. As for Palestine refugees (PR) we work for, they included all PR in their covid-19 care without discrimination. This is actually not common in humanitarian setting.

Second is the political / security risks of covid-19 infection among Syrian refugees in the camps. In some settings, covid-19 infection would negatively affect their livelihood access: e.g., work permission or movement permission. If there are such cases, people may access PCR less with the fear of losing work or movement permissions. Is there any situation like this for Syrian refugees in the camps?

Third is vaccinations. This would surely affect the severity of the covid-19 infection. If not yet, possible for you to address this in our manuscript? Again, Jordan government included refugees (at least Palestine refugees I know the best) in their national deployment and vaccination programs. This itself is a great act. Is it the same for Syrian refugees in the camps?

Again, thanks so much for your very important manuscript.

Reviewer #2: Altare et al y describe the epidemiology of COVID-19 in Azraq and Zaatari refugee camps (Jordan) and evaluate its impact on routine health services. Cases were first reported six months after the first case was reported in Jordan. Incidence rates were lower in camps than neighboring governorates and the country as a whole. Characteristics of cases and risk factors for negative disease outcomes were consistent with evidence from elsewhere. Overall health care utilization, consultations for respiratory infections, immunization and family planning declined during the first year of COVID-19, while maternal health services and non-communicable diseases were less affected. The authors report that COVID-19 transmission was lower in camps than outside of camps. Health services were affected differently, but disruptions appear to have been limited.

This is a well written and well analyzed description and the authors should be commended for their clarity and comprehensive analyses. I do not have any comments on this piece other than one key aspect. Although the authors mention, in the discussion (in fact only once on page 18 -"These results support the assertion that refugee populations living in the two largest camps in Jordan did not represent a threat of spreading SARS-COV-2 amongst the general population, as has been claimed in many countries hosting refugees" - this is in fact major in terms of the report presented here. The manuscript could be enhanced by providing additional information on this aspect as well as additional commentary from the authors. The "blaming" of refugees is both common and often egregious. I would strongly suggest to the authors to extend this discussion more and even consider being a bit stronger in the abstract, highlighting this point or consider adding a concluding paragraph to the discussion. Other than that, there are some minor typos which the authors can be ensured to address in the text. 

Reviewer #3: Comments are in the attached document. And pasted here after: 

* The article bares very little comments. It is comprehensive, quite unique in its focus, is well documented and referenced, with a rich set of conclusions which interestingly are not necessarily the intuitive ones readers would expect. 

* Authors should therefore be praised for this detailed work in collecting and processing this innovative set of evidence on refugee camps in times of COVID-19

* The article may however provide a misperception about refugees in closed camps setting as if such camps conditions constitute in themselves "counter-measures" for pandemic spread. The various biases and limitations in the interpretation of the data set are very well explained, but a reminder that the situation in these camps should not be seen as pandemic mitigation factors and generalized as such. It is obvious that this would not be the intention of the authors, but it could be a side effect of the way these conclusions are presented - without cautioning statement. A sentence or two should therefore warn against any arguments equating refugee protection condition to refugee health. It is Jordan's and UNHCR merit to ensure universal access to adequate health services to refugees in these camps, but these are still 'camps' with living conditions marked by severely restricted liberties to be no justification for protection against COVID-19. 

* Although vaccines were not available at the time of the study, the conclusions of COVID-19 being a milder problem (than expected) in refugee camp settings may be misused politically to delay access to vaccines for refugees nowadays. This may be off-subject here, but a sentence should highlight that - given the parallel in this data set with what we know of COVID-19 -vaccination is likely to hold the same benefit in refugee camps than anywhere else, and that refugee camps should remain a top priority for any national COVID-19 vaccination roll-out. 

* The impact of NPIs in camp setting, which can be implemented more easily than in open societies, is well mentioned, but would deserve more elaboration: it could be good to have (if possible) them - if not quantified (stringency index overtime) - at least better qualified, so that we can have a better understanding of their greater impact in closed camps than outside them. 

* Finally, the favorable course of the pandemic in these two camps compared to the outside communities in Jordan cannot be replicated to migrant camp settings such in Greece. This is evidently another subject, but in the commentary section, a couple of sentences contrasting the two situations could be added. Reference to: S Hargreaves, E Kondilis, D Papamichail, S McCann, M Orcutt, E Carruthers, A Veizis, The impact of COVID-19 on migrants in Greece: a retrospective analysis of national data, European Journal of Public Health, Volume 31, Issue Supplement_3, October 2021, ckab164.589, https://doi.org/10.1093/eurpub/ckab164.589 . Again, the intention is to avoid overinterpreting these conclusions to setting that may overlap but which bare very different assumptions in terms f protection, level of care, and state oversight. 

* To mention a typo in the last sentence: " there are no available data are available to analyze its effects "

* In conclusion: a great article with important data and new evidence presented, deserving immediate publication without revisions, apart from cautioning against possible misuse or over/misinterpretation of some of the conclusions at political level. 

Reviewer #4: 1. The title of the article does not fully reflect that one of the primary objectives is to assess the impact on routine health service utilization.

2. As model equation on page 4 indicates the author used a generalized linear model, what's the reason to apply the generalized additive model using mgcv function, which is usually used to capture non linearities by adding non-linear smooth functions, rather than the generalized linear model using package such as glm.nb()?

3. Since the outcome of interest is incidence/utilization rate, I would suggest to use offset(log(population)) instead of β1 log(population).

4. Are the ORs reported in Table adjusted or unadjusted? Are seasonaly pattern and secular trend taken into consideration? 

5. Page 10: "Azraq recorded an immediate decrease of 30% (IRR: 0.705, 95%CI [0.533 - 0.933]) and Zaatari of 21% (IRR: 0.794, 95%CI [0.597 - 1.056]).". Was the 30% decrease significantly different from the 21% decrease in Zaatari? The author may need to test this by using interaction. 

6. Figure 2 indicates the recovery of health utilization since the immediate decrease of 30% in outpatient visits in Azraq in Jan 2020, while no evidence of recovery in Zaatari. I would suggest to test it and present the results.

7. How did the author deal with potential outliers? Particularly in the subgroup analysis for ANC coverage, lie birth coverage and measles vaccination.

[LINK]

---

## [Decision Letter · Decision Letter 2]

13 Apr 2022

Dear Dr. Spiegel,

Thank you very much for re-submitting your manuscript "COVID-19 epidemiology and changes in health service utilization in Azraq and Zaatari refugee camps in Jordan: a retrospective cohort study" (PMEDICINE-D-22-00341R2) for review by PLOS Medicine.

I have discussed the paper with my colleagues and the academic editor and it was also seen again by three reviewers. I am pleased to say that provided the remaining editorial and production issues are dealt with we are planning to accept the paper for publication in the journal.

[LINK]

We look forward to receiving the revised manuscript by Apr 20 2022 11:59PM.   

Sincerely,

Beryne Odeny, 

PLOS Medicine

plosmedicine.org

Requests from Editors:

1. For the data availability statement - If the data are not freely available, please include an appropriate contact (web or email address) for inquiries (this cannot be a study author) 

2. Thank you for providing the STROBE checklist. Please upload the STROBE checklist as a separate supporting file (titled, “S1 Checklist”) and reference it accordingly in the Methods section. 

3. The last sentence of the Introduction needs revision as causal effects cannot be inferred, from “…and evaluate the effects of COVID-19 and its related response measures…” to “… and evaluate the associations between COVID-19 and routine health service utilization…” 

Comments from Reviewers:

Reviewer #1: Thank you so much. You kindly and nicely addressed the points i raised previously. Very clear and very well-explained!

Reviewer #2: The authors have responded to the questions posed by all reviewers. My recommendation would be to accept this manuscript for publication. 

Reviewer #4: Thank you for improving the manuscript.

Just one more minor comments:

1. The author may need to replace the term "OR (odds ratio) " by "IRR (Incidence rate ratio) " to make it consistent although they are often used interchangeably in the interpretation of the results from negative binomial models.

[LINK]

---

## [Editor Report · Decision Letter 3]

19 Apr 2022

Dear Dr Spiegel, 

On behalf of my colleagues and the Academic Editor, Dr. Rebecca Freeman Grais, I am pleased to inform you that we have agreed to publish your manuscript "COVID-19 epidemiology and changes in health service utilization in Azraq and Zaatari refugee camps in Jordan: a retrospective cohort study" (PMEDICINE-D-22-00341R3) in PLOS Medicine.

PRESS

Sincerely, 

Beryne Odeny 

PLOS Medicine